# OpenReview forum: "Adversarial Distilled Retrieval-Augmented Guarding Model for Online Malicious Intent Detection"
_ICLR.cc/2026/Conference — ICLR 2026 Conference Desk Rejected Submission_

### Official Review · Reviewer_tQKc · 2025-10-28

**Soundness:** 2
**Presentation:** 1
**Contribution:** 2
**Rating:** 2
**Confidence:** 4

**Summary:**

This paper proposes a safety-guardrail model, ADRAG which leverages document augmentation techniques to classify malicous intentions. To further improve the performance, ADRAG includes several techniques motivated by adversarial training. The proposed method show a good trade-off between classification accuracy and latency in 10 benchmark datasets.

**Strengths:**

- This paper proposes various techniques to improve the malicous intent classification while maintaining reasonable costs.

- The experiments are extensive, and the results are appealing.

**Weaknesses:**

- This paper significantly lacks clarity, and the presentation could be improved:
    - In Eq. 1, $\boldsymbol{x} = (x_1, ..., x_n)$ and $\boldsymbol{y} = (x_1, ..., x_T)$. Should $\boldsymbol{y}$ instead be $(y_1, ..., y_T)$?
    - The phrase “... is introduced in the input space” in L125 is unclear. Does it mean that the perturbation $\delta$ is applied to the input embeddings in Eq. 3?
    - The output distribution of $F_\theta$ is defined as a categorical distribution over the vocabulary in Eq. 1, but as a categorical distribution over intent classes in Eq. 5.
    - As I understand it, Eq. 5 represents the main objective of this paper. Therefore, it would be clearer to express it as $\text{arg}\max_\theta$ with the goal explicitly stated, such as $\mathcal{T}_\text{tot}(x) \leq \tau$.
    - The notation $y$ is used for intent labels after Eq. 5, but it was already used for responses in Eq. 1.

- This paper is motivated by adversarial training but lacks a proper discussion of relevant work in Section 2.

- The paper refers to a closely related work, HarmAug (Lee et al., 2024), but cites it incorrectly as Zhang et al., 2024 (L672). HarmAug (Lee et al., 2024) appears to be highly relevant, as it also aims to achieve a good trade-off between accuracy and latency. However, this paper does not include HarmAug in its experimental comparisons.

- In L384, the authors state that they used 70 NVIDIA H200 nodes. As far as I can tell, the experiments do not appear to require such extensive computational resources. Could the authors clarify the details of this usage?

**Questions:**

Please see weakness

---

### Official Review · Reviewer_9vqA · 2025-10-28

**Soundness:** 2
**Presentation:** 3
**Contribution:** 3
**Rating:** 4
**Confidence:** 4

**Summary:**

The paper presents ADRAG, a framework that integrates adversarial training, retrieval-augmented modeling, and selective knowledge distillation (SKD) for real-time malicious intent detection. The authors propose two key components:
(1) Retrieval-Augmented Adversarial Fine-Tuning (RAFT), which enhances model robustness to adversarial and noisy retrieved contexts, and
(2) Selective Knowledge Distillation (SKD), a staged distillation approach that transfers knowledge from a teacher to a smaller student model while maintaining stability and accuracy.

Empirically, the authors report that ADRAG’s distilled model (149M parameters) achieves comparable performance to large-scale guard models such as GPT-4, WildGuard-7B, and Llama-Guard-3-8B across ten safety-related benchmarks, while sustaining an end-to-end latency below 6 ms at approximately 300 QPS on a single H100 GPU.

**Strengths:**

1. The paper introduces a new approach that combines adversarial robustness training with retrieval-augmented generation (RAG) in the context of safety moderation. This integration is relatively unexplored and addresses an important practical challenge—how to make guard models resilient to noisy or adversarial retrieved contexts, which are increasingly common in retrieval-based LLM systems.
2. The authors demonstrate a well-engineered model that achieves a favorable balance between accuracy and inference speed through the proposed Selective Knowledge Distillation (SKD). The distilled 149M-parameter model attains performance comparable to large-scale guard models while maintaining real-time latency (<6 ms at 300 QPS). This positions the approach as both scientifically interesting and practically deployable for large-scale moderation systems.

**Weaknesses:**

1. Selective Knowledge Distillation (SKD) is presented as a central technical contribution of this work. However, there is no direct comparison against prior Knowledge Distillation (KD) techniques such as vanilla KD [1]. Without such ablation, it is difficult to attribute performance gains specifically to the proposed SKD mechanism. A fair comparison with established KD baselines is necessary to scientifically validate SKD as a meaningful contribution.
2. The paper describes the 395M-parameter teacher model as "high capacity," yet this is substantially smaller than competing guard models such as WildGuard (7B) [1] and Llama-Guard (8B) [2]. Given that the paper’s key contributions—adversarial fine-tuning and selective distillation—are designed to compress knowledge from large-capacity teachers, restricting experiments to a single 395M model weakens the generality of the conclusions. The study would be more convincing if it included experiments with larger teachers (≥1B parameters) to demonstrate that the proposed methods scale effectively.
3. The results rely solely on F1 score, which is highly sensitive to threshold choice and thus provides an incomplete picture of classifier robustness. Metrics such as AUPRC (Area Under Precision-Recall Curve), AUROC, or Expected Calibration Error (ECE) should be included to better evaluate real-world reliability, especially for safety moderation tasks with class imbalance.
4. All baselines used in the paper are non-retrieval models (standard guard classifiers). This is problematic given that the proposed method is a Retrieval-Augmented Guard model. To ensure fair comparison, retrieval-based baselines such as Class-RAG [3] should be included. Otherwise, the reported gains may partially result from the additional retrieval context rather than the proposed adversarial or distillation improvements.

[1]Hinton, G., Vinyals, O., & Dean, J. (2015). Distilling the knowledge in a neural network. arXiv preprint arXiv:1503.02531.
[2] Han, S., Rao, K., Ettinger, A., Jiang, L., Lin, B. Y., Lambert, N., ... & Dziri, N. (2024). Wildguard: Open one-stop moderation tools for safety risks, jailbreaks, and refusals of llms. Advances in Neural Information Processing Systems, 37, 8093-8131.
[3] Inan, H., Upasani, K., Chi, J., Rungta, R., Iyer, K., Mao, Y., ... & Khabsa, M. (2023). Llama guard: Llm-based input-output safeguard for human-ai conversations. arXiv preprint arXiv:2312.06674.
[4] Chen, J., Shen, E., Bavalatti, T., Lin, X., Wang, Y., Hu, S., ... & Jain, A. (2024). Class-rag: Real-time content moderation with retrieval augmented generation. arXiv preprint arXiv:2410.14881.

**Questions:**

1. The citation of HarmAug[1] seems incorrect in the paper. Please fix the author name.
2. Although the paper mentions HarmAug-Guard[1] and its evaluation details in the appendix A.9, the results are left out in the main table(Table 1). Please add the results of HarmAug-Guard in Table 1 for better comparison in terms of performance and model size.

[1] Lee, S., Seong, H., Lee, D. B., Kang, M., Chen, X., Wagner, D., ... & Hwang, S. J. (2024). Harmaug: Effective data augmentation for knowledge distillation of safety guard models. ICLR 2025

---

### Official Review · Reviewer_F8kq · 2025-10-30

**Soundness:** 2
**Presentation:** 1
**Contribution:** 1
**Rating:** 2
**Confidence:** 3

**Summary:**

The authors propose ADRAG, which is a two-stage system for real-time malicious intent detection that adversarially trains a retrieval-augmented teacher and selectively distills it into a 149M student that uses top-K exemplars from an evolving knowledge base for robust, context-aware, low-latency decisions. On 10 benchmarks, it matches large safety LLMs (98.5% of WildGuard-7B), outperforms GPT-4 by 3.3% and Llama-Guard-3-8B by 9.5% on OOD, and runs under 6 ms at 300 QPS; ablations show RAFT and SKD are complementary, with diminishing returns near ceiling tasks.

**Strengths:**

- The 149M student matches 98.5% of WildGuard-7B, surpasses GPT-4 by 3.3% and Llama-Guard-3-8B by 9.5% on OOD, and delivers sub-6 ms end-to-end latency at 300 QPS (up to 5.6× faster).
- It unifies retrieval-augmented adversarial training with a selective distillation scheduler and an evolving knowledge base, explicitly targeting retrieval-induced noise; thorough ablations show RAFT and SKD are complementary and effective.

**Weaknesses:**

###  Notation
- In equation (1), there is no $\theta$ in the left-hand side. The probability distribution must be parameterized by a model with some parameters $\theta$.
- The notation for label is inconsistent. In equation (1), the label $\mathbf{y}$ is a sequence of tokens. However, in equation (7) and training dataset $\mathcal{D}$,  the label $y_i$ looks like a single token (or class).

- The same letter $T$ is used for multiple different purposes. $T$ denotes  maximum sequence length, teacher, or training epoch.

- The same epsilon is used in both equation (6) and (8), which implies that the two sets are disjoint. But one is obtained by perturbing the other a little bit, so they should not be disjoint.

- The symbol $\hat{p}_{y_i}$ is not defined.

- Equation (15) doesn't seem to make sense. $y_i$ is not defined. It literally says that confidence becomes 1 when all the prediction agrees. Otherwise it is zero.

### Method
- Derivation of the proposed adversarial perturbation is not properly grounded. What is distance metric to measure the perturbation $\delta$? How the proposed adversarial perturbation can maximize the negative log-likelihood while enforcing the perturbation norm less than $\rho$?

- Retrieving relevant examples from knowledge base looks critical. I am not sure using off-the-shelf sentence encoder works for this specific task.

### Experiments
- The comparison with baselines might not be fair because they do not use examples for in-context learning. It would be better to include the same examples in the baselines for a fair comparison.

- As written in related work section, HarmAug seems to be very relevant baseline which distill Llama-Guard-3 model for efficient deployment. But it is missing in the experiments.

- The effectiveness of using examples from knowledge base is questionable. There is no ablation study on retrieval part.

**Questions:**

Please see the weaknesses above.

---

### Official Review · Reviewer_kXdy · 2025-11-03

**Soundness:** 2
**Presentation:** 3
**Contribution:** 2
**Rating:** 6
**Confidence:** 4

**Summary:**

The paper proposes ADRAG—a framework for online malicious‑intent detection that combines Retrieval‑Augmented Adversarial Fine‑Tuning (RAFT) on a large teacher with Selective Knowledge Distillation (SKD) into a compact student, plus an Evolving Knowledge Base (EKB) for continual, context‑aware inference. The teacher is trained to be robust to perturbations of both the user query and retrieved contexts (few-shot examples); the student inherits this behavior via a scheduler that alternates teacher‑only, co‑distillation, and student‑only phases. On ten ID/OOD safety benchmarks, a 149M student attains ~98.5% of WildGuard‑7B's performance and outperforms GPT‑4 and Llama‑Guard‑3‑8B on average OOD F1, while keeping p99 < 6 ms end‑to‑end latency at 300 QPS (model + retrieval).

**Strengths:**

- The proposed framework is both novel and interesting. In particular, the online evolving knowledge base enables continuous model updates using user feedback and policy-driven synthetic examples.
- The distilled 149M model achieves strong performance with low latency.
- The paper presents detailed results and ablation studies.

**Weaknesses:**

- The effectiveness of the proposed method remains questionable. Despite a rather complex training pipeline, the final results show only modest improvements: the average score increases from 0.8765 to 0.8910 on the ID datasets, and from 0.8416 to 0.8614 on the OOD datasets, compared to a standard SFT BERT model. Furthermore, the baseline BERT model is evaluated in a zero-shot setting, while the proposed methods utilize few-shot examples. As a result, the added complexity and cost of the training pipeline are difficult to justify.

- Some important ablation studies are missing. For example:

    - The Context Perturbation module contains many components, but it is unclear whether all of them are necessary.

    - The selective knowledge distillation method adds extra complexity. How does it compare to standard distillation?

    - What would happen if we simply distilled a large Guardrail model into BERT?

- The online evolving knowledge module is conceptually interesting, but I could not find any related experiments. For instance, does it enable better continual learning?

**Questions:**

- How are the components in the Context Perturbation module selected? Have you performed any ablation studies on them?

- What is the performance when distilling a RAFT-trained model directly using a standard distillation objective?

- Have you compared your approach with a simple distilled model or other distillation methods? I noticed that HarmAug is mentioned in the appendix, but I couldn't find its results.

- Have you evaluated the evolving knowledge base in your experiments?

---

### Note · Program_Chairs · 2026-01-17
**Submission Desk Rejected by Program Chairs**

The following references in this submission do not refer to real documents and/or have major errors in bibliographic information:

 Xinyu Zhang et al. Safety chain-of-thought: Aligning language models by training them to decompose safety violations. arXiv preprint arXiv:2403.00012, 2024b. URL https://arxiv.org/ abs/2403.0001